# Combined Exposure to Birch Pollen and Thunderstorms Affects Respiratory Health in Stockholm, Sweden—A Time Series Analysis

**DOI:** 10.3390/ijerph19105852

**Published:** 2022-05-11

**Authors:** Mare Lõhmus, Tomas Lind, Laura MacLachlan, Agneta Ekebom, Björn Gedda, Pia Östensson, Antonios Georgelis

**Affiliations:** 1Institute of Environmental Medicine, Karolinska Institutet, Nobels väg 13, 171 77 Solna, Sweden; tomas.lind@sll.se (T.L.); laura.maclachlan@regionstockholm.se (L.M.); antonios.georgelis@regionstockholm.se (A.G.); 2Centre for Occupational and Environmental Medicine, Region Stockholm, Solnavägen 4, 113 65 Stockholm, Sweden; 3Department of Environmental Research and Monitoring, Swedish Museum of Natural History, 104 05 Stockholm, Sweden; agneta.ekebom@nrm.se (A.E.); bjorn.gedda@nrm.se (B.G.); pia.ostensson@nrm.se (P.Ö.)

**Keywords:** birch pollen, thunderstorms, respiratory health

## Abstract

Background: Thunderstorm asthma is a term used to describe surges in acute respiratory illnesses following a thunderstorm and is often attributed to an intense exposure to aeroallergens. Several episodes of thunderstorm asthma have been observed worldwide; however, no such cases have been described in Sweden. In Sweden, the most prominent exposure to air-borne pollen occurs during the blooming of the birch. We aimed to explore the associations between respiratory health and the combined exposure to thunderstorms and birch pollen. Methods: We investigated the association between the daily numbers of outpatient visits due to respiratory cases and the combined exposure to thunderstorms and birch pollen during the period of 1 May–31 September in 2001–2017, in Stockholm County, Sweden, by using time series analysis with log linear models. Results: We detected noticeable increases in the number of outpatient visits on both the same day (max 26%; 95% CI 1.16–1.37) and the day after (max 50%; 95% CI 1.32–1.70) the occurrence of a thunderstorm, when the concentrations of birch pollen and the number of lightning discharges were within the highest categories. Conclusions: It is possible that co-exposure to heavy thunderstorms and high concentrations of birch pollen affects the respiratory health of the Stockholm population. To the best of our knowledge, this is the first study addressing the thunderstorm-related respiratory illnesses in Sweden and the effects of birch pollen. Our study may be important for future public health advice related to thunderstorm asthma.

## 1. Introduction

‘Thunderstorm asthma’ refers to an increase in cases of acute respiratory illness following the occurrence of thunderstorms in the local vicinity [1]. The name of this phenomenon is, however, slightly misleading, as not all cases of thunderstorm asthma have a previous asthma diagnosis and also non-asthmatic individuals are known to experience bronchoconstriction during these events [2]. According to Idrose et al., by the year 2020 a total of 22 outbreaks of epidemic thunderstorm asthma had been registered worldwide since the early 1980s, with most of these occurring in Australia and the United Kingdom (UK) [3].

Although the mechanisms of thunderstorm asthma are not entirely understood, these events are thought to be caused by exposure to airborne allergenic particles (i.e., pollen grains and fungal spores) concentrated in thunderstorm downdrafts [4]. The occurrence of thunderstorm asthma in Australia has mostly been associated with high levels of grass pollen, but studies from the UK have suggested that other types of pollen, such as tree and weed, may also be important triggers [5]. Whilst whole pollen grains are too large (>20 μm) to penetrate deep into the human airways and give rise to thunderstorm-related bronchoconstrictions, sub-pollen particles, released when the pollen grains rupture under certain conditions, are smaller (<3 μm) and capable of travelling into the small airways [4]. The reasons that pollen ruptures during thunderstorms are not fully understood, but studies have suggested a combination of factors, such as mechanical friction from wind gusts, lightning activity within thunderstorm clouds and water-induced swelling. Sub-pollen particles are then concentrated and transported to ground level by cold downdrafts or outflows from a thunderstorm [4]. The large number of these small-size pollen granules in the downdrafts is what is thought to give rise to the asthma symptoms and respiratory distress during thunderstorm asthma events [3]. Pollen may, however, not be the only cause of thunderstorm asthma, and other aeroallergens, such as fungal spores, have been suggested as potential triggers [6,7,8].

A systematic review by Idrose et al. [3] found nine observational case-report studies about thunderstorm asthma, six of which were conducted in Australia and three in the UK. Most of these studies focused on grass pollen as the primary exposure [9,10,11,12,13,14,15], while a few [9,13,15] mentioned the influence of other kinds of pollen as well. In addition, Idrose et al.’s review [3] included eleven publications with time-series analyses from the UK [8,16,17,18,19,20], Australia [6,21,22] and Canada [23], which explored the association between thunderstorms and asthma admissions while taking into account pollen levels. Several of these studies [17,18,19,22] found positive associations between the lagged effects of pollen and thunderstorm asthma, while some [6,8,20,21,22,23,24] found little to no association between pollen exposure, thunderstorms and asthma admissions. To best of our knowledge, no studies of the kind have been conducted in Sweden or in the rest of the northwestern Europe.

Pollen from birch and other related trees of the families Betulaceae and Fagaceae are considered to be the top-ranked producers of aero-allergenic pollen in northwestern Europe, and the major cause of allergic rhinitis and possibly seasonal asthma symptoms [25,26,27,28,29]. In addition, the considerable cross-reactivity and sequential pollen season of the tree species with related types of allergens expose many sensitized individuals to a prolonged period of allergic symptoms [26,30,31,32]. Consequently, the negative impact on the quality of life of individuals allergic to birch pollen and the subsequent clinical importance of birch-pollen-related health effects is substantial [26,33].

In the present study, we explored potential associations between the daily number of outpatient visits resulting from a spectrum of respiratory symptoms and the combined exposure to thunderstorms and birch pollen during the period 1 May–31 September over the years 2001–2017, in Stockholm County, Sweden, by using a time series analysis with log linear models. We used a wider selection of respiratory diagnoses than most previous studies as the health problems arising during the thunderstorm asthma events may include a variety of diffuse symptoms and may not be limited to asthma. Based on previous studies, we expected to find a positive association between pollen exposure, thunderstorms and outpatient visits due to respiratory causes. The findings of our investigation are important for discovering the potential associational patterns between thunderstorms and respiratory illnesses, which are today widely unrecognized in Sweden. Furthermore, since such sudden surges of acute respiratory illness may suddenly increase the need for acute health services and deplete available resources, our results may contribute to increasing the preparedness of the Swedish medical care system.

## 2. Methods

### 2.1. Data Collection

#### 2.1.1. Outpatient Visits Due to Diseases in the Respiratory System

We investigated the association between respiratory health problems, thunderstorms and wind-borne birch pollen concentration in Stockholm County, Sweden, during the period 1 May–31 September (153 days) over the years 2001–2017. Anonymized data of all outpatient visits in Stockholm County due to respiratory illness (ICD l0 diagnoses: J09–J18 J20–J22 J40–J44 J47 J45) were obtained from the National Patient Register [34] at the National Board of Health and Welfare (Socialstyrelsen). In total 737,377 outpatient visits resulting from the above-defined respiratory causes occurred during the study period. Outpatient visits in Sweden do not require a stay in hospital and involve a meeting between the patient and healthcare staff. They can be both planned and unplanned (acute).

#### 2.1.2. Weather

Thunderstorms very rarely occur in Stockholm County before and after the period 1 May–31 September. Therefore, our analysis was concentrated on this period of the year only. Records of thunderstorm days and the number of lightning discharges per day over the years 2001–2017 were provided by the Swedish Meteorological and Hydrological Institute (SMHI). The SMHI calculates the discharge data using information collected by automatic flash location sensors. The thunderstorm data were used either as continuous variables or divided into five categories: cat 0 = 0 days without thunderstorms; cat 1 = 1–13 discharges (≤50th percentile); cat 2 = 14–90 discharges (>50th percentile to ≤75th percentile); cat 3 = 91–249 discharges (>75th percentile to ≤90th percentile); cat 4 = 250–5881 discharges (>90th percentile). In addition, information about hourly temperatures, humidity, air pressure, and wind direction and speed, were downloaded from the open information sources of SMHI (https://www.smhi.se/data/meteorologi accessed on 1 February 2020).

#### 2.1.3. Pollen and Air Pollution

Birch pollen sampling was performed by specialists at the Palynological laboratory, The Swedish Museum of Natural History, Stockholm [35], using the Burkard volumetric pollen and spore trap based on the Hirst design [36]. Airborne particles were deposited on sticky tape mounted on a drum, which was slowly turned by clockwork. The tape was embedded in stained glycerine gelatine and analyzed under an optical light microscope using × 400 magnification. The pollen of different taxa were determined and their numbers were counted according to the 12 transversal transects method [37]. The sampling method produced the pollen count expressed as the concentration in pollen grains/m^3^/24 h. By using this method, pollen data have been collected in Stockholm since 1973. The pollen trap is located on the roof of a building about 15 m above ground level at Stockholm University Campus (59°21′56.2″ N 18°3′34.7″ E).

The daily concentration of ambient birch pollen was used either as a continuous variable or divided into five categories based on quartiles of the pollen concentration levels (cat 0 = 0 days without birch pollen; cat 1 = 1 grain/m^3^; cat 2 = 2–6 grains/m^3^; cat 3 = 17–39 grains/m^3^; cat 4 = 41–4759 grains/m^3^).

Exposure data for air pollution were provided by the Stockholm-Uppsala County Air Quality Management Association and the Swedish Environmental Research Institute. Hourly means of Nitrogen oxides (NO_x_) were obtained from roof-level monitors within central Stockholm.

### 2.2. Statistical Analyses

#### 2.2.1. Confounders

The number of healthcare visits due to respiratory illness in Stockholm County will also vary as a result of variables other than our main exposures (daily birch pollen concentrations and the number of lightning discharges). For example, the day of the week and occurrence of public holidays, as well as various long-term trends in the frequency of patient visits (due to i.e., population increase), may affect the likelihood of a person seeking help for respiratory health problems. Thus, our analyses had to be adjusted for both day-specific differences and for between-years trends.

Since the risk of exposure for both birch pollen and thunderstorms varies during the 153 day study period, the “season time” itself becomes a confounder. To be able to adjust our models for the seasonal time trends, we explored two different approaches. In the first approach (models 1 and 2), we first numbered all days in the data set (153 days/year) regardless of year. Thereafter, we adjusted our models for time trends by using sinus and cosinus terms with wavelengths at 3, 4, 6, 12, and 24 months (see Figure 1), as well as a linear term (the day number) and a quadratic term (the day number squared). In the second approach (model 3), we instead used 80 dummy variables (one for each study month) when adjusting for season time.

Several meteorological variables are likely to influence the association between pollen exposure and respiratory health [38]. According to previous studies, daily (natural cause) mortality in Stockholm is the lowest around an average temperature of 12 °C [39,40]. Thus, when adjusting our models for the daily average outdoor temperature, we used a piecewise linear spline with one knot at 12 °C. In addition, our models were adjusted for daily average values of relative humidity (using a restricted cubic spline model with three knots), wind speed, and air pressure. Furthermore, the daily average levels of NO_x_ at the baseline were included in the models.

#### 2.2.2. Models

All statistical analyses were performed using Stata 14.2 for Windows (StataCorp LLC, College Station, TX, USA). We conducted the descriptive between-group analyses (Table 1 and Table 2) by either using a *t*-test (in case of two groups only) or one-way-ANOVA (in case of more than two groups). To explore the potential associations between the daily numbers of outpatient visits due to respiratory cases and the combined exposure to thunderstorm-associated lightning discharges and birch pollen, we used time series analysis with log linear models. In addition, we obtained post-regression estimates by using the “lincom” command (linear combinations of parameters in Stata 14.2).

Model 1a addressed the association between the daily number of outpatient visits from respiratory causes, birch pollen concentrations (divided into five categories) and thunderstorm-associated lightning discharges (divided into five categories) on the same day. The model was adjusted for seasonal time trends (sinus–cosinus, linear and quadratic terms), as well as the daily average estimates of outdoor temperature, humidity, wind speed, air pressure and NOx. In addition, sensitivity analyses (models 1b and 1c) were performed to address the more specific associations between our main exposures and the outcome. Model 1b estimated the associations between daily number of outpatient visits and the birch pollen concentration (here as a continuous variable) on days when the number of lightning discharges was within the two highest categories (cat 3 and cat 4). Model 1c investigates the relationship between the daily numbers of outpatient visits and the lightning discharges (here a continuous variable) on days with birch pollen concentrations within the two highest quartiles (cat 3 and cat 4).

Models 2a, 2b and 2c correspond to Models 1a, 1b and 1c respectively; however, instead of investigating the exposure–outcome associations on the same day, models 2a, 2b and 2c associate the number of outpatient visits to the exposure variables on the day before the outpatient visit. In addition, model 3 investigates associations between outcome and exposure on the day before the outpatient visit; however, it uses a different, more rigorous approach regarding the adjustment for season time. Instead of using the adjustment by sinus–cosinus, linear and quadratic terms, in model 3 these variables are replaced by 80 dummy variables (the months May–September during 2002–2017 were included in the model).

## 3. Results

The background characteristics of the study sample are shown in Table 1 and Table 2. During the total study period of 2448 days, lightning discharges were registered on 799 days and birch pollen on 993 days. Simultaneous exposure to thunder and birch pollen occurred on 272 days.

The number of lightning discharges per day was higher at high relative humidity (compared to low relative humidity), at high ambient temperature (compared to low ambient temperature) and at low air pressure (compared to high air pressure) (Table 1). Concentrations of NO_x_ and wind speed were not significantly associated with the number of lightning discharges/day. The concentrations of birch pollen were likely to be higher at low relative humidity (than at high relative humidity), at high air pressure (than at low air pressure) and at low ambient temperature (than at high temperature) (Table 1). Wind speed showed a tendency to be negatively related to birch pollen concentrations while the concentrations of NO_x_ were not associated with birch pollen.

The number of outpatient visits from respiratory causes increased through the years 2002–2017 (Figure 1 and Appendix A). The average number of outpatient visits was highest on Tuesdays and lowest on Saturdays (Table 2). No obvious pattern could be detected when examining the association between the number of lightning discharges and the number of outpatient visits; however, there seemed to be a positive relationship between the number of outpatient visits and birch pollen concentrations, as well as between the number of outpatient visits and wind speed, air pressure and NOx (Table 2). The relationship between ambient temperature and the number of outpatient visits was negative (Table 2).

### 3.1. Associations between Outcome and Exposure Estimates on the Same Day

Model 1a estimated the association between the daily number of outpatient visits from respiratory causes, birch pollen concentration (categorized on five levels) and thunderstorm-associated lightning discharges (categorized on five levels) on the same day. The results for the full model can be found in the Appendix A. The post-estimation results are shown in the Table 3. We found several significant interactions between the levels of birch pollen and the number of lightning discharges (Appendix A). A significant increase in the number of outpatient visits was detected at high ambient birch pollen levels combined with a high number of lightning discharges (Table 3).

Results from the sensitivity analyses, where model 1b (the association between the daily number of outpatient visits and birch pollen concentrations measured on a continuous scale on days with lightning discharges within categories 3 and 4) showed that the number of outpatient visits increased by 3% per 100 pollen grains/m^3^/day (95% CI 1.011; 1.048, *p* < 0.001). Model 1c (association between the daily numbers of outpatient visits and the number of lightning discharges on days with birch pollen concentrations within the two highest quartiles) found no significant association between the outcome and the number of lightning discharges (a 3% increase in the number of outpatient visits per 100 lightning discharges/day (95% CI 0.989; 1.006, *p* < 0. 534)).

### 3.2. Associations between Outcome and Exposure Estimates on the Day Before

Model 2a investigated the association between the number of outpatient visits and the exposure variables, namely birch pollen concentration and the number of lightning discharges, on the day before the outpatient visit. The results of the full model can be found in the Appendix A. Results from the post-estimation analyses are shown in Table 4. As in model 1a, we found significant interactions between the levels of birch pollen and the number of lightning discharges (Appendix A), as well as a significant increase in the number of outpatient visits following a day with high ambient birch pollen levels combined with a high number of lightning discharges (Table 3).

The first sensitivity analysis, model 2b, investigated the association between the daily numbers of outpatient visits and birch pollen concentrations measured on the day before the patient visit on a continuous scale, when the number of lightning discharges was within categories 3 and 4. The model found a non-significant positive association between the levels of birch pollen and the number of outpatient visits (1.018; 95% CI 0.995; 1.041; *p* < 0.132). The second sensitivity analysis, model 2c, addressed the association between the daily numbers of outpatient visits and the number of lightning discharges on the day before the outpatient visit, when birch pollen concentrations were within the two highest quartiles. Model 2c found a non-significant positive association between the number of outpatient visits and the number of lightning discharges (1.001; 95% CI 0.995; 1.007, *p* < 0. 845).

Model 3 also investigated the association between the number of outpatient visits and the exposure variables on the day before the outpatient visit but used a different kind of adjustment for the time and season trends. The results for model 3 can be found in Appendix A, while Table 5 shows the results from the post-estimation analysis. According to Table 5, this model indicated a rather large increase (up to 50%) in the number of outpatient visits following a day with a high birch pollen concentration and a thunderstorm with a high number of lightning discharges compared to the previous models.

## 4. Discussion

In the present study, we analyzed the association between outpatient visits from respiratory causes, birch pollen concentrations and the number of lightning discharges over time in Stockholm County. We found that the number of outpatient visits from various respiratory causes increased significantly on both the same day and the day after a thunderstorm event, when either the concentration of ambient birch pollen or the number of lightning discharges was within the highest quartile. When both the concentrations of ambient birch pollen and the number of lightning discharges were within the two highest categories, the number of outpatient visits due to respiratory causes increased by up to 26% on the same day as the thunderstorm event and up to 40–50% (depending on the model) on the day after the thunderstorm event compared to the days without ambient birch pollen.

Several previous studies [9,10,11,12,13,14,15,17,18,19,22] have investigated the roles of pollen and thunderstorms in asthma admissions. Most of these have concentrated on specific epidemic asthma events following a thunderstorm, while others, similarly to the present study, analyzed long-term associations between asthma admissions and the combined exposure of pollen and thunderstorms. Newson et al. [19], for example, investigated the roles of thunderstorms and grass pollen counts as predictors of daily hospital admissions for asthma in England. They found that thunderstorms with exceptionally high densities of sferics (lightning flashes) combined with high grass pollen counts up to four days before the medical care visit were associated with a relative excess risk of around 25% for asthma admissions in both children and adults. Silver et al. [22] found an interactive effect between exposure to thunderstorms and grass pollen concentrations and the number of daily asthma admissions in Melbourne, Australia. However, while increased grass pollen levels were positively and significantly associated with increased asthma admission rates at lag three days, the thunderstorms by themselves were not associated with asthma admission rates.

Our findings partly agree with previously reported results. For example, like Newson et al. [19], we observed an interactive effect of ambient pollen and the number of lightning discharges only when the levels of both exposures were high. Additionally, like Silver et al. [22], the results from our sensitivity analyses suggested that the number of respiratory care visits is more directly associated to ambient pollen concentrations than to the occurrence of thunder storms. For example, we found that the association between the daily numbers of outpatient visits and birch pollen concentration measured on a continuous scale on days with lightning discharges within the two highest categories was statistically significant and that the number of outpatient visits increased by about three percent per 100 pollen grains/m^3^/day. The association between the daily numbers of outpatient visits and the number of lightning discharges measured on a continuous scale on days with birch pollen concentrations within the two highest quartiles was, however, not significant.

There were several differences in our study design compared to most previous publications. The majority of the previous reports investigating associations between asthma admissions, pollen and thunderstorms have concentrated on the role of grass pollen exposure [9,10,11,12,13,14,15]. Since birch is the major producer of allergenic pollen in northern Europe, and also the most common cause of allergy-related symptoms in Sweden and Scandinavia [26,32,41], in the present study we opted instead to investigate the role of birch pollen. However, we also performed a few diagnostic analyses using the grass pollen data and saw no tendencies for an interactive effect of grass pollen and lightning discharges on the number of outpatient visits (data not shown).

The main outcome in most previous studies concerning the episodes of thunderstorm asthma was the number of asthma admissions [3]. In the present paper, we used a broader range of respiratory diagnoses, since some reports indicate that more diffuse symptoms than those associated with asthma have been observed and that individuals without an asthma diagnosis have been affected [1,12]. As long-term data from emergency hospital visits in Stockholm County were not available, we used data from outpatient visits. Outpatient visits in Sweden involve a meeting between the patient and healthcare staff and can be either planned or acute. Our data did not differentiate between planned and acute visits, which is a shortcoming. Significant or unexpected changes in the number of outpatient visits are, however, more likely to arise due to an increased number of acute rather than planned visits.

To best of our knowledge, this the first study that has investigated the interactive effects of thunderstorms and pollen on respiratory health in Sweden and in Scandinavia. According to our findings, it is possible that the interactive effect of high ambient birch pollen concentrations and heavy thunderstorms may increase the occurrence of respiratory health problems in Stockholm County. In general, the phenomenon of thunderstorm asthma is very poorly known in Sweden, both among the general public and medical caregivers. However, epidemic thunderstorm asthma is thought to be a global phenomenon, which can strike rather unexpectedly with catastrophic consequences, meaning it should be prepared for [42]. Furthermore, because of climate change, it is likely that the incidence of thunderstorm asthma will increase and have wide-reaching, negative consequences [42].

Early warning systems designed to prevent loss of life and avoid surge events on healthcare infrastructure during future epidemic incidences of thunderstorm asthma have been introduced in a small number of locations globally. For example, in 2017 the Victorian Government in Australia launched an online thunderstorm asthma alert system that was designed to help the general public be better prepared for such an occurrence [43]. This web-based information site teaches people about what epidemic thunderstorm asthma is, how they should prepare for it and also how they should make use of the forecast. It is, however, also necessary to identify individuals that are potentially at-risk of respiratory health problems during a thunderstorm asthma event, to be able to direct the necessary information in a more precise way.

Lee et al. [12] examined patient admission data following the largest thunderstorm asthma epidemic ever reported—November 2017 in Melbourne, Australia—with the goal of identifying the key susceptibility factors. The authors reviewed the records of all patients aged 16 years and older with symptoms suggestive of asthma (dyspnea, respiratory distress, cough and wheeze) and found that only 40% had a known asthma diagnosis prior to admission. However, almost all of the admitted patients suffered from allergic rhinitis during the grass pollen season and were ryegrass-pollen-sensitized. Furthermore, 94% of the cases had been exposed to the thunderstorm during time outdoors. The data available for the present study did not include information about previous patient history regarding birch pollen sensitization and allergic rhinitis, meaning they did not allow us to investigate these potential susceptibility factors.

## 5. Conclusions

To conclude, it is possible that co-exposure to heavy thunderstorms and high concentrations of birch pollen affects the respiratory health of the population in Stockholm County. More studies from other locations in Sweden and from rest of the Northern Europe are needed to be able to draw more definite conclusions. Nevertheless, experience from other countries could be used to raise an important public health message recommending that individuals who are sensitized to pollen stay indoors when there is a risk of thunderstorm. Furthermore, proper medication use by asthma patients may be important in the prevention of acute healthcare visit during combined pollen and thunderstorm events.

## Figures and Tables

**Figure 1 ijerph-19-05852-f001:**
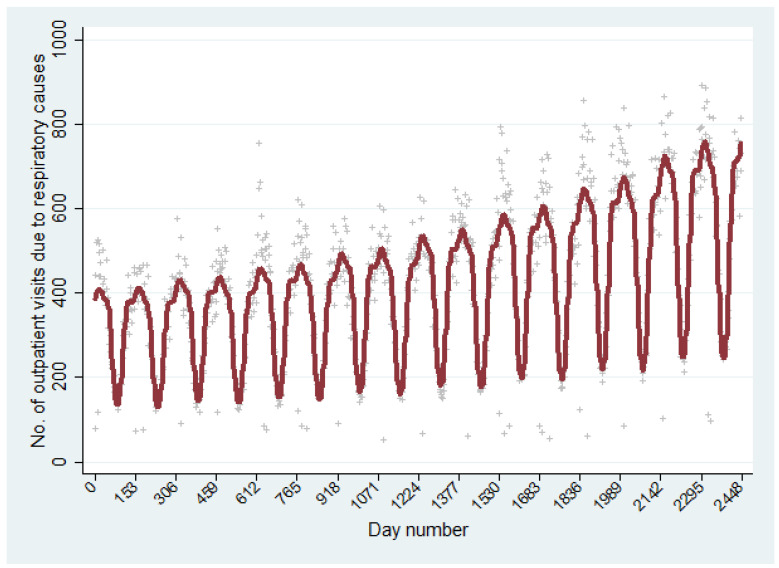
GLM model showing the number of outpatient visits due respiratory causes over time. The days of the total observation period (153 days per year, ranging between 1 May and 31 September annually from 2002 to 2017) are coded as days 1–2448, regardless of the year. The adjustment for time trends was performed by using sinus and cosinus terms with wavelengths at 3, 4, 6, 12, and 24 months, as well as a linear term (the day number) and a quadratic term (the day number squared).

**Table 1 ijerph-19-05852-t001:** Relationship between the main and co-exposures. The *p* values express statistical significance between sub-categories of each variable at the 95% CI level.

Daily Average Estimates of the Co-Exposures	Lightning Discharges (*n*/day)		Birch Pollen Grains (*n*/m^3^/day)	
	Mean (95% CI)	*p*<	Mean (95% CI)	*p*<
**Relative humidity (%)**				
**32.79–72.25**	19.81 (12.88; 26.75)	0.0001	66.18 (48.78; 83.59)	0.0001
**72.29–99.88**	74.18 (54.65; 93.72)		10.56 (6.8; 14.32)	
**Wind speed (m/s)**				
**0.17–2.67**	53.41 (37.16; 69.67)	0.2172	45.64 (29.64; 61.64)	0.0875
**2.68–7.33**	40.33 (27.41; 53.24)		29.98 (22.04; 37.92)	
**Air pressure (hPa)**				
**985.43–1012.64**	60.01 (46.19; 73.84)	0.0121	23.79 (16.02; 31.55)	0.0015
**1012.64–1040.05**	33.53 (18.11; 48.94)		52.9 (36.79; 69.00)	
**NOx (μg/m^3^)**				
**1.32–11.90**	42.59 (29.41; 55.77)	0.4441	28.88 (20.67; 37.09)	0.1286
**11.92–78.08**	50.79 (34.43; 67.15)		41.99 (27.18; 56.81)	
**Outdoor temperature (°C)**				
**1.37–15.32**	20.4 (14.27; 26.53)	0.0001	62.77 (47.69; 77.85)	0.0001
**15.33–26.05**	73.18 (53.47; 92.89)		14.49 (4.75; 24.23)	

**Table 2 ijerph-19-05852-t002:** Relationship between the number of outpatient visits due to respiratory causes per day and the daily average estimates of the exposure variables. Note: *p*-values express statistical significance between sub-categories of each variable at the 95% CI level.

Exposure Variables	n (days)	No. of Outpatient Visits Mean (95% CI)	*p*<
**Year ***	2448	301.22 (292.71; 309.72)	0.000
**Weekday**			
**Monday**	350	447.19 (444.98; 449.41)	0.000
**Tuesday**	350	462.88 (460.63; 465.14)	
**Wednesday**	349	411.93 (409.8; 414.07)	
**Thursday**	350	389.33 (387.26; 391.4)	
**Friday**	350	244.44 (242.81; 246.09)	
**Saturday**	349	73.58 (72.68; 74.48)	
**Sunday**	350	78.83 (77.9; 79.76)	
**Lightning discharges (*n*/day)**			
**cat 0: 0**	1649	320.94 (320.08; 321.81)	0.000
**cat 1: 1–13**	407	260.87 (259.31; 262.45)	
**cat 2: 14–90**	192	274.29 (271.95; 276.64)	
**cat 3: 91–249**	97	277.27 (273.96; 280.6)	
**cat 4: ≥250**	103	217.57 (214.73; 220.44)	
**Birch pollen grain/m^3^**			
**cat 0: 0**	1435	267.42 (266.58; 268.27)	0.000
**cat 1: 1**	273	287.78 (285.77; 289.8)	
**cat 2: 2–6**	245	335.99 (333.7; 338.29)	
**cat 3: 17–39**	229	388.62 (386.07; 391.18)	
**cat 4: 41–4759**	246	390.18 (387.71; 392.66)	
**Relative humidity (%)**			
**32.79–72.25**	1222	301.5 (289.41; 313.6)	0.889
**72.29–99.88**	1217	300.29 (288.26; 312.32)	
**Wind speed (m/s)**			
**0.17–2.67**	1229	290.96 (279.36; 302.55)	0.019
**2.68–7.33**	1210	311.37 (298.9; 323.83)	
**Air pressure (hPa)**			
**985.43–1012.64**	1224	292.58 (280.85; 304.31)	0.046
**1012.64–1040.05**	1224	309.85 (297.53; 322.17)	
**NOx (μg/m^3^)**			
**1.32–11.90**	1202	231.93 (220.34; 243.52)	0.000
**11.92–78.08**	1201	367.46 (356.12; 378.8)	
**Outdoor temperature (°C)**			
**1.37–15.32**	1225	357.81 (344.77; 370.86)	0.000
**15.33–26.05**	1223	244.52 (234.56; 254.49)	

* The year-average values are shown in Appendix A.

**Table 3 ijerph-19-05852-t003:** The change (%) in the number of daily outpatient visits due to respiratory causes related to the birch pollen concentrations (pollen grains/m^3^/day; in five categories) and the number of lightning discharges per day (in five categories) on the same day. The post-estimation β coefficient values, including the confidence interval (CI) values at the 95% significance level, are depicted below the change estimate (%). Significant results are marked with *.

Birch Pollen on the Same Day	No. of Lightning Discharges on the Same Day				
	Cat 0	Cat 1	Cat 2	Cat 3	Cat 4
pollen cat 1	REF	1%	−1%	7%	13%
		1.01 (0.91–1.11)	0.99 (0.82–1.20)	1.07 (0.97–1.19)	1.13 (1.03–1.24) *
pollen cat 2	REF	1%	0%	8%	14%
		1.01 (0.93–1.11)	1.00 (0.83–1.20)	1.08 (0.99–1.18)	1.14 (1.05–1.24) *
pollen cat 3	REF	10%	−9%	2%	10%
		1.10 (1.03–1.18) *	0.91 (0.69–1.20)	1.02 (0.95–1.10)	1.10 (1.03–1.17) *
pollen cat 4	REF	−2%	11%	25%	26%
		0.98 (0.86–1.12)	1.11 (1.01–1.23) *	1.25 (1.16–1.35) *	1.26 (1.16–1.37) *

**Table 4 ijerph-19-05852-t004:** The change (%) in the number of daily outpatient visits due to respiratory causes related to the birch pollen concentrations (pollen grains/m^3^/day; in five categories) and the number of lightning discharges per day (in five categories) on the day before. The post-estimation β coefficient values, including the confidence interval (CI) values at the 95% significance level, are depicted below the change estimate (%). Significant results are marked with *. Adjustment for the season time was performed using sinus–cosinus, linear and quadratic terms.

Birch Pollen on the Day before	No. of Lightning Discharges on the Day before Outpatient Visit				
	Cat 0	Cat 1	Cat 2	Cat 3	Cat 4
pollen cat 1	REF	−4%	1%	6%	9%
		0.96 (0.80–1.16)	1.01 (0.77–1.32)	1.06 (0.95–1.19)	1.09 (1.00–1.19) *
pollen cat 2	REF	−3%	4%	5%	11%
		0.97 (0.86–1.05)	1.04 (0.98–1.10)	1.05 (0.97–1.14)	1.11 (0.98–1.25)
pollen cat 3	REF	7%	17%	3%	14%
		1.07 (0.91–1.25)	1.17 (1.03–1.33) *	1.03 (0.82–1.28)	1.14 (1.07–1.22) *
pollen cat 4	REF	−4%	26%	40%	27%
		0.96 (0.78–1.18)	1.26 (1.15–1.38) *	1.40 (1.30–1.51) *	1.27 (1.15–1.40) *

**Table 5 ijerph-19-05852-t005:** The changes (%) in the number of daily outpatient visits due to respiratory causes related to the birch pollen concentrations (pollen grains/m^3^/day; in five categories) and the number of lightning discharges per day (in five categories) on the day before. The post-estimation β coefficient values, including the confidence interval (CI) values at the 95% significance level, are depicted below the change estimates (%). Significant results are marked with *. Adjustment for the season time was performed using dummy variables.

Birch Pollen on the Day before	No. of Lightning Discharges on the Day before Outpatient Visit				
	Cat 0	Cat 1	Cat 2	Cat 3	Cat 4
pollen cat 1	REF	9%	18%	36%	45%
		1.09 (0.91–1.30)	1.18 (0.87–1.59)	1.36 (1.15–1.61) *	1.45 (1.21–1.72) *
pollen cat 2	REF	3%	15%	20%	17%
		1.03 (0.87–1.21)	1.15 (0.91–1.46)	1.20 (1.06–1.36) *	1.17 (1.02–1.35) *
pollen cat 3	REF	15%	41%	13%	35%
		1.15 (0.98–1.35) *	1.41 (1.21–1.63) *	1.13 (0.90–1.43)	1.35 (1.20–1.53) *
pollen cat 4	REF	7%	47%	50%	42%
		1.07 (0.84–1.36)	1.47 (1.29–1.67) *	1.50 (1.32–1.70) *	1.42 (1.18–1.72) *

## Data Availability

Restrictions apply to the availability of these data. The outcome data was obtained from National Patient Register at the National Board of Health and Welfare (Socialstyrelsen) and are available at https://www.socialstyrelsen.se/en/statistics-and-data/registers/national-patient-register/ with the permission of the Swedish Ethical Review Authority and the National Board of Health and Welfare.

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
