# Peer review of "Combined Exposure to Birch Pollen and Thunderstorms Affects Respiratory Health in Stockholm, Sweden—A Time Series Analysis"

_ijerph, 2022, doi:10.3390/ijerph19105852_

Round 1

Reviewer 1 Report

This is a well-written manuscript that describes a time series analysis looking at the association between respiratory health and both birch pollen exposure and thunderstorms. The authors report that associations between respiratory ill-health and these exposures were detected when outpatient visits occurred on the same day or the day after the occurrence of thunderstorms. They  conclude that it is possible that such combined exposures may affect respiratory health in Stockholm.  The authors need to clarify the following points:

  1. The raw data presented in Table 2 suggests that the number of outpatient visits due to respiratory symptoms is higher when there are no lighting discharges which seems to be opposite to the hypothesis that thunderstorms and birch pollen exposures should be associated with increased visits. Why is this?
  2. Whilst it is conceivable that such exposures to birch pollen and thunderstorms may increase the incidence of respiratory systems, it seem unlikely that such an effect would be detected on the same day of thunderstorms, given that there must be a finite time between the actual thunderstorm, the generation of the pollen particles (the presumed risk factor), the development of symptoms and presentation at the health care facility. Is this really conceivable?
  3. Figure 1 seems to indicate that the number of outpatient visits due to respiratory causes has increased over the time period from 2002-2017. Why is this? Does it represent a simple increase in target population or changes in population sensitivity (e.g. due to changes in ethnicity).  Or does it represent changes in diagnostic criteria or access to actual healthcare?  If so, what impact would this have on their results
  4. The authors indicate that they used admissions due to  “respiratory symptoms” rather than asthma as the outcome measure. Have they not repeated their analyses using the more specific definition of asthma symptoms as presumably people with asthma would be more susceptible to developing further problems.
  5. How were the thunderstorm categories decided upon?
  6. How representative is measuring birch pollen at apparently one location on the Stockholm University Campus to that exposure occurring at different locations in Stockholm County? Similarly, with thunderstorms occurring potentially throughout the County at different locations and at different times, this may well result in significant exposure misclassification. Please explain how this might affect the results.

Author Response

This is a well-written manuscript that describes a time series analysis looking at the association between respiratory health and both birch pollen exposure and thunderstorms. The authors report that associations between respiratory ill-health and these exposures were detected when outpatient visits occurred on the same day or the day after the occurrence of thunderstorms. They  conclude that it is possible that such combined exposures may affect respiratory health in Stockholm.  The authors need to clarify the following points:

Thank you very much for your efforts of reviewing this manuscript and for you valuable comments!

1. The raw data presented in Table 2 suggests that the number of outpatient visits due to respiratory symptoms is higher when there are no lighting discharges which seems to be opposite to the hypothesis that thunderstorms and birch pollen exposures should be associated with increased visits. Why is this?

The yearly observation period of 153 days in this manuscript, starts on May 1st and ends on September 31st. The thunderstorms mainly occur from May to August and the birch pollen is around in May and in the beginning of June. Due to the Swedish tradition of long summer holidays, leading to less medical care visits during the summer, it is very common that many doctors’ appointments take place after the holidays, in September, which is a month with very few thunderstorms and no ambient birch pollen. This could be an explanation to why we found higher number of doctoral visits in the raw data on days without lightning. We did our best to adjust our models for such time trends.

2. Whilst it is conceivable that such exposures to birch pollen and thunderstorms may increase the incidence of respiratory systems, it seem unlikely that such an effect would be detected on the same day of thunderstorms, given that there must be a finite time between the actual thunderstorm, the generation of the pollen particles (the presumed risk factor), the development of symptoms and presentation at the health care facility. Is this really conceivable?

Episodes of acute thunderstorm asthma epidemics have been characterized by a very rapid increase in the number of asthma emergency room visits (within 30h) during and following the hours of a thunderstorm. As our data included both emergency and planned visits, we chose to explore the effect on both the same day and the day after.

3. Figure 1 seems to indicate that the number of outpatient visits due to respiratory causes has increased over the time period from 2002-2017. Why is this? Does it represent a simple increase in target population or changes in population sensitivity (e.g. due to changes in ethnicity).  Or does it represent changes in diagnostic criteria or access to actual healthcare?  If so, what impact would this have on their results

The figure most likely illustrates the general population growth in Stockholm County. Stockholm is one of the fastest growing cities in Europe and between 2002 and 2017 the population of Stockholm County increased from 1.8 million to 2.3 million

4. The authors indicate that they used admissions due to  “respiratory symptoms” rather than asthma as the outcome measure. Have they not repeated their analyses using the more specific definition of asthma symptoms as presumably people with asthma would be more susceptible to developing further problems.

We absolutely agree with the reviewer that it had been important to pay specific attention to people with asthma; unfortunately, the data about previous asthma diagnoses was not available for us. We also used a wider selection of respiratory diagnoses since the health problems arising during the thunderstorm asthma events often include a variety of diffuse symptoms, which are not always diagnosed as “asthma”.

5. How were the thunderstorm categories decided upon?

(cat 0=days without thunderstorms; cat 1=1-13 discharges (≤ 50th percentile); cat 2=14-90 discharges (> 50th percentile; ≤ 75th percentile); cat 3=91-249 discharges (>75th percentile; ≤ 90th percentile); cat 4=250-5881 discharges (> 90th percentile)). We have now added this description to the paper.

6. How representative is measuring birch pollen at apparently one location on the Stockholm University Campus to that exposure occurring at different locations in Stockholm County? Similarly, with thunderstorms occurring potentially throughout the County at different locations and at different times, this may well result in significant exposure misclassification. Please explain how this might affect the results.

We agree with the referee that our pollen estimates are simply an indication of the ambient pollen concentrations and may not reflect the levels locally. According to some previous measurements, when several rooftop pollen traps were used by the pollen laboratory in Stockholm, the ambient levels of pollen were very similar between the traps, despite them being several kilometers apart. Thus, even if it is sadly impossible to estimate how much pollen exactly a person was exposed to before he/she visited the health care provider, the pollen data available for us did let us identify the periods of time with “high” and “low” levels of ambient pollen in the Stockholm County. Since we did not have any data about person-specific movements, and thus could not identify in what part of the city the person was exposed to the pollen or the lightning, we could, in this case, not be able to use more detailed geographical information.

Reviewer 2 Report

The manuscript “Combined exposure to birch pollen and thunderstorms affects respiratory health in Stockholm, Sweden – a time series analysis” is an interesting environmental study of possible mechanisms of thunderstorm induced respiratory illness exacerbations in northwestern Europe. The study is well planned, described and results are nicely presented. I have learned some new facts about the possible impact of environmental factors on obstructive lung disease course. The study would be perfect if the authors would be able to select strictly the asthma associated outpatient visits not the whole set of outpatient visits in planned and acute stays in hospital. However, even with wide-ranging analysis the study is interesting and beneficial for the reader.

Author Response

The manuscript “Combined exposure to birch pollen and thunderstorms affects respiratory health in Stockholm, Sweden – a time series analysis” is an interesting environmental study of possible mechanisms of thunderstorm induced respiratory illness exacerbations in northwestern Europe. The study is well planned, described and results are nicely presented. I have learned some new facts about the possible impact of environmental factors on obstructive lung disease course. The study would be perfect if the authors would be able to select strictly the asthma associated outpatient visits not the whole set of outpatient visits in planned and acute stays in hospital. However, even with wide-ranging analysis the study is interesting and beneficial for the reader.

We would like to thank the referee for the comments, and we absolutely agree that a data set with acute medical care visits only would have been preferable. As we stated in the manuscript, we also used a wider selection of respiratory diagnoses since the health problems arising during the thunderstorm asthma events may include a variety of diffuse symptoms, which are not always diagnosed as “asthma”. Furthermore, the statistical power of the study had been rather insufficient if only asthma diagnoses had been used. Thank you for your efforts of commenting this manuscript.

Reviewer 3 Report

It was a pleasure to review this paper; I have no concerns with the content whatsoever.

Minor comments: please replace all instances of lighting with lightning, and if you mean raise, you should not write rise (l. 46 and54). In line 344 you mean to say differentiate, I would think; in line 352 you write exasperate, which does not fit.

In the conclusion you mention that sensitized patients should have antihistamine supplies; you could also mention that sensitized patients should know whether or not they have asthma, and if so, need to make sure they use their medication as precribed. Antihistamines by themselves would be unlikely to prevent a respiratory-related healthcare visit.

Author Response

It was a pleasure to review this paper; I have no concerns with the content whatsoever. Minor comments: please replace all instances of lighting with lightning, and if you mean raise, you should not write rise (l. 46 and54). In line 344 you mean to say differentiate, I would think; in line 352 you write exasperate, which does not fit.

Thank you very much for your comments and for righting our spelling mistakes! I believe that we have made all suggested changes!

In the conclusion you mention that sensitized patients should have antihistamine supplies; you could also mention that sensitized patients should know whether or not they have asthma, and if so, need to make sure they use their medication as prescribed. Antihistamines by themselves would be unlikely to prevent a respiratory-related healthcare visit.

Thank you, this is, of course completely correct! We have now changed the end part of the manuscript to:

Nevertheless, experience from other countries could be used to raise an important public health message recommending that individuals who are sensitized to pollen stay indoors when there is a risk of thunderstorm. Furthermore, proper medication use by asthma patients may be important in the prevention of acute healthcare visit during combined pollen and thunderstorm events.